# Empirical and Heuristic Phenomenological Case Study of the HeartMath Global Coherence Initiative

**DOI:** 10.3390/ijerph16071245

**Published:** 2019-04-08

**Authors:** Stephen D. Edwards

**Affiliations:** Psychology Department, University of Zululand, Private Bag X1001, KwaDlangezwa 3886, South Africa; sdedward@telkomsa.net

**Keywords:** empirical, heuristic phenomenology, case study, HeartMath, Global Coherence Initiative

## Abstract

Along with the creativity of vast technological advances, humanity’s endemic destructiveness continues. Planetary healing needs motivated this research. The aim was an empirical and heuristic phenomenological investigation into and an evaluation of the theoretical and technological implications of the HeartMath Global Coherence Initiative. The single case study, and limited amount of data, indicated the null hypothesis. Methodology included HeartMath Inner Balance tool and newly developed Global Coherence application (app). Data collection involved linked empirical measures and experiential journaling. Quantitative data analysis, which consisted of statistical analysis of correlations between six existing Global Coherence magnetometers and empirical measures of meditation records, from Inner Balance and Global coherence apps, respectively, yielded unexpected findings, both significant and insignificant, in the form of trends towards global and local group coherence, respectively. Qualitative findings essentially revealed variations on the, interrelated, consciousness themes of wholeness, holistic healing, energy healing and meditation. In addition to various limitations and implications, interpretation of integrative findings indicated theoretical and practical support for the HeartMath mission and vision of developing and promoting personal, social and global coherence.

## 1. Introduction

A given phenomenology for meditators, global coherence may appear as an interconnectedness behind the passing moments and events of ordinary conscious awareness [1,2]. For example, interconnected consciousness facilitated the Buddha teaching *pratītyasamutpāda,* interdependent origination in a “world woven of interconnected threads” [3]. Holistic consciousness enabled Jesus Christ to affirm “before Abraham was I am” [4]. More recently, coherent consciousness facilitated Einstein’s recognition of space-time [5], Bohm’s [6] notion of an implicate order and Sheldrake’s [7] morphogenetic fields. Such evidence has provided support for the postulation of an interconnecting global information network facilitated through the Earth’s magnetic field [8,9,10,11]. Space aviation, artificial intelligence technology, the internet and smartphones are further contemporary practical examples of such global coherence. 

Along with the creativity of vast technological advances, humanity’s endemic destructiveness regularly features in contemporary international news, in the form of chaos, incoherence, corruption, crime, injustice, international terrorism and other general violence [12]. Planetary threats of global warming and shocking natural disasters exacerbate the general impression of a planet crying out for healing. Nowhere is this more apparent than in poorer Asian and African regions where overpopulation, inequity, unemployment, poverty and illness reflect typically human, endemic struggles between survival and destruction, coherence and incoherence, flourishing and floundering [13]. However as Richo [14] notes, givens such as suffering also bring humanity’s greatest gifts, such as joy and compassion. Nowhere are these truisms more apparent than in beneficial human, social relationships, as portrayed in the Zulu theme of Ubuntu and German term “*mitwelt*”. It is in this existential, human, social, relational, “with world” or “we world” that human relationships are forged, begin, flourish or flounder, and end. 

Along with beneficial social relationships and coherent communication, optimal consciousness is vital for health and wellbeing. For instance, in wisdom traditions, sages such as Buddha, Christ and Mohammed proposed that the great value of meditation is its propensity to facilitate consciousness, particularly moral consciousness, behaviour, creativity and health promotion [15,16]. Rosch [17] has extolled the value of biological, electromagnetic and subtle energy medicine as mind-matter interface. Well-controlled, collective consciousness studies, using time series methodologies, have indicated significant correlations between size of meditation group, reductions in war deaths and/or intensity, as well as improvement in broad quality of life indices [18]. Various meditative practices, especially centred on the integral heart, positive and renewing emotions such as love, care, compassion and appreciation, led Doc Childre, to create the HeartMath system in 1991 [19].

The HeartMath system pursues a central vision and mission of interdisciplinary, integral, heart focused research towards promoting personal, social and global coherence and health. Major findings relate to heart communication of electromagnetic, neurochemical, biophysical and hormonal information [10,20]. Pribram’s [21] holonomic, dynamic, systemic, pattern recognition research provided scientific theory for much early HeartMath research. From a practical perspective, heart rate variability (HRV) clearly emerged as key index of communication, adaptation, resilience and general health, reflecting autonomic nervous system (ANS) dynamics, heart-brain and other organ system interactions, as well as resonant interconnections with broader energetic environment, geomagnetic field, Schumann resonances, solar activity and cosmic rays [10,20,22].

This particular article is inspired by the Institute of HeartMath, Global Coherence Initiative (GCI) pioneering research that continues to support converging scientific evidence for vast, energetic, interconnectivity at human, planetary and solar systemic levels. Coherence is a key concept, which includes resonance, synchronization and interconnectedness ranging from physical and psychophysiological modes through personal and social coherence, to global interconnectivity [19]. The Global Coherence Initiative (GCI), launched in 2008, refers to a global network of ultrasensitive magnetic field detectors. In addition to resonating with the phenomenological insights of the ancient sages, this contemporary dynamic systemic approach is scientifically grounded on considerable evidence available on websites; Heartmath.org and http://www.glcoherence.org.

In this context, health is understood in terms of diverse arrays of dynamic energy including physiological systems, behavioural patterns and environmental resonant frequencies [19]. Various environmental health studies provide examples. McCraty, et al. [23]. Found significant HRV correlations with solar and geomagnetic activity as well as geomagnetic field-line and Schumann resonances over a longitudinal 31-day period in a group of participants in separate locations. Timofejeva, et al. [24] found synchronization between slow HRV wave rhythms and changes in local magnetic field data, with the degree of synchronization being associated with the quality of interpersonal relationships. Extending earlier studies, Alabdulgader, et al. [25] as well as McCraty et al. [26] again indicated that energetic environmental phenomena impact psychophysical processes in people in different ways depending on their sensitivity, health status and capacity for self-regulation.

To date no study has specifically addressed global coherence meditation experiences with particular reference to HeartMath Global Coherence measurements. The aim of this empirical, phenomenological case study was to address this particular gap in the HeartMath research literature. Analogous to a tidal wave in Piquet being the result of a butterfly flapping its wings in Tibet, it was recognized that any single person case study was highly unlikely to be significant quantitatively. Thus the null hypothesis was invoked in this respect. However, it was considered that any empirical data could provide support for the communication of meaning that might be derived from an experiential investigation and evaluation into the theoretical and technological implications of the Global Coherence Initiative and newly developed Global Coherence app.

## 2. Materials and Methods

### 2.1. Instruments

Detailed information on the two instruments used in this study, the Inner Balance trainer and Global Coherence apps are available on the internet and HeartMath website at www.HeartMath.org. Briefly Inner Balance measures, monitors and provides psychophysiological coherence biofeedback. The Global Coherence app has additional functions. It measures, monitors and provides psychophysiological coherence, group and global biofeedback. More details follow in the research procedure as well as results and discussion sections.

### 2.2. Ethical Considerations

The study followed ethical standards in accordance with the Declaration of Helsinki. Institutional approval was obtained from the Zululand University research committee, project number S894/97. The author is a registered clinical, educational, sport and exercise psychologist and licensed HeartMath coach and mentor. This was considered necessary for measurement and intervention purposes and did not indicate conflict of interest as the focus of the HeartMath Institute is to promote personal, social and global health. Certainly, from a critical, reflexive perspective, the pilot study will have been influenced by the authors’ knowledge and experience of HeartMath techniques, tools and interventions. The many limitations of the present study are readily acknowledged. Phenomena such as the placebo, Hawthorne effect and general expectancy and relationship variables will all have featured.

### 2.3. Theoretical Framework

This study is contextualized within a holistic, integral, theoretical framework, as advocated by such philosophers as Smuts [27], Aurobindo [28] and Wilber [29]. Phenomenology has been described as critical theory, movement, approach, method, technique and intervention, typically representing consciousness from interior, subjective perspectives [30]. The approach used in the present heuristic phenomenological study is based on the work of Moustakas [31] who grounded his initial work on his experiences with his daughter. The present case study included a review of the author’s personal and global coherence experiences during using HeartMath, Inner Balance and Global Coherence apps over two months, with the author functioning as both researcher and participant [31]. Although such phenomenological case studies have disadvantages such as subjectivity, bias and generalizability issues, they have advantages of uniqueness, flexibility and in depth exploration that may illuminate theory and stimulate further exploration [32,33,34].

### 2.4. Case Study

This particular pilot type, case study is heuristic and phenomenological in research methodology as well as practical investigation, with the author using imaginative variation in meditation methods [31]. Such autobiographical approaches are subject to qualitative research criteria such as authenticity, faithfulness, integrity, credibility, dependability and transferability [34]. They are transparent and scientific to the extent to which they convey authentic reflexivity and are exposed to proper critical scrutiny as in any peer journal review system The Inner Balance training journal enhanced systematic, scientific objectivity with reference to HeartMath global coherence theory, practice, measurement and evaluation. The study is integrative in its concern with both quantitative HeartMath measurements and qualitative, heuristic, phenomenological descriptions of coherence experiences associated with the author’s personal praxis using the Inner Balance and HeartMath global coherence apps [35]. He used HeartMath technology to monitor or provide coherence biofeedback on all meditation sessions. In this context, it should be mentioned that the term “meditation” is used here in a generic sense to describe various explorations and interventions in consciousness, which can include contemplation and prayer, as well as reflexive thought, divergent, convergent and integral.

### 2.5. Phenomenological Bracketing

This typically requires an initial bracketing of bias (epoche) which includes transparent statement of personal assumptions and biases. The author is a 69-year-old, formerly retired, Emeritus Professor of Psychology and Research Fellow, who is happily married with two children and three grandchildren, with a fourth on the way. He has travelled widely, with health promotion presentations in over 30 countries. He is deeply grateful for an academic career that facilitated promotion of human rights and produced many postgraduate, masters and doctoral students, most of whom lived in rural Zululand, who have become professors, chairpersons of the South African national psychology association and served society and promoted health in many other ways. In addition to community psychological services and travel, he is also deeply grateful to have retirement time to teach, research and practice what he loves and where he can contribute. His meditation experiential descriptions are biased in terms of inclination to what could be described as psychological, philosophical, spiritual or mystical. Although he was brought up in Christian tradition and his basic meditation practice is Christian orientated, e.g., Centering Prayer and Prayer of the Heart, he appreciates and eclectically practices many wisdom traditions, especially African ancestral consciousness. For example, he uses essentially similar practices belonging to Christian, Muslim, Hindu, Buddhist, and Taoist practices, which respectively reflect 34%, 24%, 15%, 6.5%, and 0.3% of the planet’s human population beliefs [36]. He regularly uses HeartMath technology to monitor or provide coherence biofeedback on meditation sessions.

The author’s involvement in this study may be understood in terms of three historical streams of evidence. The first was after he discovered similarities between HeartMath methods and an African heart-breath workshop developed around the concept of Shiso, an ancient isiZulu respectful (*hlonipha*) term for a human being [37]. SHISO became an acronym for a particular breath and heart based healing method, standing for Spirit (*Umoya*), Heart (*Inhlizyo*), Image (*Umcabango),* Soul (*Umphefumulo*) and Oneness (*Ubunye*). The workshop takes the form of five steps, one for each letter of the acronym [38,39]. The second was in subsequent research collaboration with HeartMath Research Director, Dr. Rollin McCraty, and establishment of the African Global Coherence Monitoring at Hluhluwe, in KwaZulu-Natal, South Africa. The third stream consisted of the authors training as HeartMath Coach/Mentor and related research studies, some of which appear in the HeartMath research library.

### 2.6. Research Procedure

Research procedure consisted of three phases. The first retrospective phase began during a HeartMath 21 day challenge, issued to all HeartMath coach/mentors by Alan Strydom, Director HeartMath South Africa, of meditating for 21 min for at least 21 days. In response, the author recorded his coherence measurements and journaled his experiences during 74 sessions of Inner Balance practice that lasted for 21 min as well as 79 other sessions of varying length. All 153 entries were accompanied by a description of some meditation experience as recorded in the Inner Balance journal or training log. The second phase consisted of precisely correlating these meditation sessions with global coherence data in the Schumann Resonances region of the magnetic field, available in the form of live data on the website https://www.heartmath.org/gci/gcms/live-data/gcms-magnetometer. Here six site magnetometers in California, Saudi Arabia, Lithuania, Canada, New Zealand and South Africa provided frequency data from 0.32 to 36 Hertz, in the form of 24-h moving averages plotted for each site and updated hourly. The third prospective phase consisted of recording of personal praxis experiences of using the HeartMath Global Coherence app, which was in beta phase testing at the time and thus undergoing continuous refinements and improvements, to which this study could contribute. As was the case in the HeartMath 21 day challenge, the author joined a South African group for global coherence practice. The group, with a total of five members, called itself “South AfriCAN Coherence,” with the mission of “Uplifting South Africa through individual, social and global coherence”. It should also be mentioned that at this stage, the author was aware of preliminary, empirical findings of the first retrospective phase that indicated correlations between his personal meditation practice and total global coherence as well as on three individual magnetometers, but not with the local African magnetometer. So this prospective phase was accompanied by some intention to redress this perceived imbalance with respect to more local and group “South AfriCAN” focus.

### 2.7. Data Collection 

Data collection continued for just over two months, from 1 January to 4 March 2019. Empirical and experiential data collected together consisted of linked quantitative and qualitative records for each meditation session. Quantitive data consisted of coherence measurements, qualitative data were experiences from meditation. Qualitative data initially consisted of the abovementioned recorded experiences from 153 Inner Balance sessions. Further qualitative data in the form of recording and analyzing global coherence experiences on the Global Coherence app reached data saturation point and report readiness after 64 recorded sessions in 38 days coherence practice with the app. At this stage the author’s individual coherence points recorded were 59,221; the group coherence points were 74,614; and global coherence points were 3,057,300. This global coherence practice was valuable from a qualitative perspective, in terms of experiences, and greatly complemented the Inner Balance app in this regard. Finally 100 of the most meaningful experiences from 100 meditation sessions, 74 from the Personal Balance app and 26 from the Global Coherence app praxis, were selected for qualitative analysis.

### 2.8. Data Analysis 

Data analysis consisted of three phases, quantitative, qualitative and integrative. Quantitative data were analyzed with Statistical Package for the Social Sciences (SPSS, version 25, IBM, London, England) for basic descriptive statistics and Spearman correlations for non-parametric data. The conventional probability symbol of one (*) and two asterisks (**) refer to statistical comparisons, significant at levels of *p* < 0.05 and *p* < 0.01 respectively. Qualitative data were subjected to three layers of analysis, each involving increasing depth of interpretation. Repeated reading of all journal experiences for enhanced reflexivity and sense of the whole was complemented by NVivo content analysis as a form of course sieve. This analysis was done in the company of another researcher with whom the author was working collaboratively. The second level involved allocating the 100 selected experiential responses from meditation sessions into nine broad categories representing mental, physical and spiritual dimensions of personal and social, which is an adaptation of Wilber’s all quadrants all levels (AQAL) type design [29,40,41]. Ninety nine percent agreement was reached with a second collaborative researcher. Most entries reflect the author’s life orientation that Wilber, Patten, Leonard and Morelli [42] have termed Integral Life Practice (ILP), which is a holistic, synchronistic attempt at exercising body, mind and spirit in self, culture and nature. The third phase consisted of further intensive reading and reflexive resonating of experience in the sense of a double hermeneutic, which was followed by recording of essential themes of consciousness that emerged. This third phase was absolutely crucial in order to allow thick description and deeper meaning to emerge. Finally quantitative and qualitative findings were integrated, discussed, evaluated and recommendations made.

## 3. Results and Discussion

Quantitative findings follow in the form of main findings and related discussion of descriptive and correlational statistics. Qualitative findings are followed by qualitative data on the patterns and essential summary of experiences with special reference to regional and global coherence.

### 3.1. Quantitative Findings

These are indicated in Table 1 in the form of statistics for coherence and achievement scores and in Table 2 in terms of Spearman correlation coefficients for coherence and achievement scores as well as various global coherence measurements.

From Table 1 it can be observed that mean Inner Balance coherence level score of 5 and mean achievement score of 701 were found. These mean scores and their respective standard deviations and ranges are regarded as typical of this particular participant. His highest coherence and achievement scores to date are 8.6 and 8121 respectively, with a previous calculation indicating a mean coherence score of 7.53 from a cluster of the highest ten scores, 4.8 from a cluster of median 10 scores and 0.99 from the lowest cluster of 10 frequency scores [43].

Table 2 refers to Spearman correlation matrix from the individual participant’s coherence and achievement scores as recorded on the Inner Balance app when compared to global coherence scores as recorded on the six different magnetometer sites (GC101 to GCI06), as well as total global coherence (GCITOT) score derived from the sum of all 6 magnetometer scores. Empirical findings in Table 2 should be interpreted with extreme caution. The member of measures (153 sessions) is very small and only reflects average coherence level and achievement level of one person using a HeartMath Personal Balance app practice in South Africa. However, the following findings deserve some discussion. Firstly, correlations between coherence and achievement are negative, which is explainable in that the higher achievement scores reflect longer sessions and the longer a session, the more likely a lower coherence score will accrue. This negative and positive correlational pattern generally continues in relation to correlations with different magnetometers and total global coherence, where significant correlations were found with the magnetometers, GCI02, GCI03, GCI05 and total global coherence GCITOT. There were also a significant negative correlation (−0.206 *) as well as a significant positive correlation (0.367 **) between the participant’s respective coherence and achievement score and total global coherence (GCTOT). Moreover, GC106 magnetometer site has the strongest positive correlation with GCITOT (0.777 **).

In view of the fact that the participant was living closest (about 300 km away) to the South African magnetometer (GCI06), it is interesting that neither his individual coherence (0.020) nor achievement (−0.022) scores were significantly correlated with this local magnetometer. Amongst infinite other possibilities, these findings may have been associated with the extreme heat, lightning and related erratic magnetic activity over the entire Schumann Resonances spectrum during peak summer months in this region. As would be expected GCITOT is significantly correlated, negatively or positively, with coherence, achievement and all other individual magnetometer correlations. Thus, although the table is generally indicative of a pattern of interconnectedness, further investigation in randomized controlled trails with large numbers of participants is essential.

The highly significant positive correlation (0.367 **) between the participant’s achievement score and total global coherence (GCTOT) resonates with a study by McCraty et al. [26] with 20 participants, who wore 24-h ambulatory HRV monitors for two weeks. Results indicated that a Heart Lock-In technique of 15 min has a strong influence on the relationship between cardiac and geomagnetic activity. The longer meditation sessions in the present study were very similar to the Heart Lock-In technique. Certainly the present author fully endorses the view that such meditation techniques are associated with better health conditions, as correlated with heart rate variability and geomagnetic activity.

In concluding discussion in this section, it needs mentioning that the insignificant findings in relation to correlations between the individual participant and GCI06 scores are exactly opposite of what could be expected in view of the considerable personal meaning that the GCI 06 magnetometer at Bonamanzi Game Park has for the author and his family. Following the request to assist with establishment of an African global coherence magnetometer, Bonamanzi was chosen above three other possible sites, as the most suitable. This was made possible by personal family relationship with the owner of the park. A week before the installation of the magnetometer in June 2015, the author’s son had been married in the park and at his speech to the married couple after the ceremony, the author mentioned the synchronicity of these occasions. A video of the South African GC106 installation at Bonamanzi may be found at the following link: https://www.heartmath.org/gci/gcms/heartmath-south-africa-global-coherence-monitoring-system-installation/. Although too few, there remain a group of committed people in HeartMath South Africa, at the University of Zululand and at Bonamanzi Game Park who are dedicated and motivated to promote coherence and health, as well as act on their intentions as was the case recently in repairing the battery to render GCI06 fully operational.

Table 3 refers to Spearman correlation matrix from the individual participant’s coherence and achievement scores as recorded on the Global Coherence app when compared to global coherence scores as recorded on the six different magnetometer sites (GC101 to GCI06), as well as total global coherence (GCITOT) score derived from the sum of all 6 magnetometer scores. Table 3 findings should be interpreted with even more extreme caution than Table 2 as they only reflect 55 meditation sessions on the Global Coherence app, with a relatively low individual mean coherence level of 4.36 and standard deviation of 1.25. Like Table 2, coherence was negatively correlated with GCI06. However, in the case of Table 3, it should be noted that coherence is more strongly correlated with GC106 than any other magnetometer and that this correlation is significant (−0.305 *). Also, as was the case with Table 2, GCI06 has the highest positive correlation with GCITOT (0.882 **). Unlike Table 2, only coherence scores, not achievement scores, are reflected. For research purposes, it seems important that future developmental features of the Global Coherence app include readily accessible achievement scores results for individual sessions.

If the line of argument, begun with regard to Table 2, that negative correlations between coherence and achievement are related to session length, is continued for Table 3, it seems likely that readily available achievement scores would have been positively correlated with local and global sites. By extension, the hypothesis emerges that the longer an individual meditation session, the more likely, and stronger, will be the correlations between local/group and global coherence. This hypothesis is aligned with the postulation of a noosphere or cognitive, consciousness sphere encircling the earth that has emerged through evolution as a consequence of evolving complexity/consciousness [11,44].

While maintaining an extremely cautious interpretative stance empirically, Table 2 and Table 3 findings, taken together as trends or patterns, do provide some preliminary validation of the Global Coherence app. Thus, it can be very tentatively interpreted, that the still to be fully developed Global Coherence app does seem to differentially measure local or group as well as global coherence. It also seems likely that a clearer pattern may have emerged if the participant was wearing continuous cardiac monitoring equipment, as in the study by Timofejeva el al. [24]. The following qualitative findings also reflect this local and global connection and provide some theoretical support for the HeartMath mission and vision of sequentially developing the three inextricably interrelated spheres of personal, social and global coherence.

### 3.2. Qualitative Findings

The following NVivo Word Cloud in Figure 1 illustrates an initial, holistic “course sieve” of qualitative findings.

Repeated reading and reflexivity clarified the overall pattern of the whole, facilitating coding of experiential data into nine categories of consciousness representing personal, social and global coherence in physical, mental and spiritual dimensions as indicated in Table 3. Experiential sessions were inextricably interrelated and interwoven, so these are simply very loose and convenient categories for further illustrative purposes.

The distribution of the 100 entries in the nine dimensions is self-evident in Table 4. In addition to a convenient coding system, this theory driven, thematic content analysis facilitated the following, essential summary of the 100 experiential responses. (In passing, it should be mentioned, that Table 4 and Table 5, that follows, are convenient illustrations of ordered reflexivity for instructional, audit trail and summary purposes of the full data set of 100 journaled experiences of meditation sessions). A selected few follow.

#### 3.2.1. Global Coherence Essential Summary

Repeated reflexive reading further clarified the overall pattern of the whole of this study in its essential concern with consciousness, healing, or making whole and transforming from illness to health, particularly through the use of energy healing and other meditation methods. These themes are reflected in experiential journaling of meditation sessions as follows. The themes have been edited for language and sources, which have been added for greater reader friendliness and meaning. It needs passing mention that the philosophical and poetic quotes reflect both the author’s inclinations as well as their propensity to describe, amplify and elaborate experiential meaning. This is also required by such qualitative research criteria as truthfulness, credibility, faithfulness, integrity, accuracy, and dependability, in experiential and scientific fields with no final answers.

##### Holistic Healing Consciousness

For want of a better one, the tautological phrase “holistic healing” is used to convey the ongoing experience of the whole, or of healing as whole making event, occurring moment to moment. This is also an evolutionary process, connecting continually towards greater wholeness as distinct from simultaneous processes of entropy and dissolution. This process is illustrated by the spinning global coherence symbol as coherence waxes and wanes, and spinning Taoist symbol depicting oscillating yin yang patterns. Although one cannot absolutize or reify any single aspect of global coherence, a central fundamental essence is its intervention towards interconnectedness, wholeness and healing, especially through heart, love, compassion (100 Global Mental). For example, one journal entry (91 Global Mental) reads:

Listening to the synchronized sounds of singing silence and shushing sea, like spirit and soul, and all great values such as love, truth, justice and freedom. All are holographic, universal and individual expressions of each other in a symphony of spheres. A Zulu Ubuntu idiom concerns hand washing ‘*isandla sigezesinye*’ which literally means one hand is unable to wash itself [45]. Ultimately all may reflect great global, holistic movements of evolution and involution, dispersion and integration, interconnectedness and interdependence, from and to a theoretical omega point or origin of all reality according to de Chardin [44]. There is oneness, non-duality, and a supreme synthesis in global coherence. Jung [13] drew mandalas. Dante [46] spoke of the Love that moves the sun and other stars. T.S. Elliot [47] said of the eternal moment: “We must not cease from exploration and the end of all our exploring will be to arrive where we began and to know the place for the first time”. Wordsworth’s poetry is beautiful. He sang of “a motion and a spirit that impels all thinking things, all objects of all thought, and rolls through all things”. Reflexivity indicates that all are attempts to express the ultimately inexpressible. Neither words nor numbers nor anything else can express this: as Lao Tse might put it: “The way that can be named is not the eternal Way” [48]. Or as Popper [49] would put it, as postulates in their potential to reveal further truth, hypotheses may be rejected or supported, never verified or proven.

##### Energy Healing Consciousness

This is a subtheme of both holistic healing, and meditation. It is an energetic activity occurring through awareness, consciousness, intentionality, physical activity, person-world interrelationships, social relations, and naturally existing environmental interconnectedness. Although apparent in all realms, physical, mental and spiritual, the participant seems to consciously practice energy healing most at spiritual level as apparent in following three spiritual quotes in personal, social and global dimensions:

“Consciously practising release or kenosis. Relaxing any feeling of tension, especially in jaw area. Experience of free (God, motion, spirit, chi) energy flooding through body system. Cortisol peak release or vagal break” (7 Personal Spiritual).

“We dance around a ring and suppose, while spirit sits in the middle and knows”. Praxeological, cultural, scientific, understanding of experienced subtle energy bodies as mapped by Judith [50] Assagioli [51] and Wilber [29], providing predictive frames. “Nina, my daughter and shadow, interests in yoga and personal growth with Bali visit for yoga conference. Alan Strydom will also be there.” (18 Social Spiritual).

“Consciously circulating subtle energetic life and light, releasing and radiating from inner to outer world. Initially felt as ecstatic light, the conscious radiating event requires concentrative effort at times interspersed with continuing relaxing to surf coherence waves. Thankfully extreme heat and humidity of past few days is over. Southern wind today will bring cooling rain.” (44 Global Spiritual).

##### Meditation Involving Specific Exploration and/or Interventions in Consciousness

In addition to their specificity with regard to meditation activity and related reflexivity, the following entries describe meditation activity in greater depth and detail:

“Silent, stillness, awareness. Apophatic kenosis. Conscious releasing, shedding, jettisoning of tension, clenches, stiffness, body armour, like a snake shedding its skin. Persona, ego, self-transcendence ongoing, opening like a flower opening to the sun, a soul opening to spirit in a journey into that which has always been here and actually impossible to avoid. Amazing how one misses this so often in everyday life hustle and bustle. In such moments of silent truth, greater reality reveals itself. This is why meditation is so valuable. It reminds one to recognize truth in itself, to suspend assumptions, thoughts, in fact everything. It is also a natural phenomenon and process that one should not get too intense about. Rather simply enjoy the ride. The moment arrives soon enough. There may not be a reason for doing it until after it has been done, as one ultimately reaches a greater place of consciousness that solves any previous problems and issues that one may have had with the process or experienced generally.” (25 Global Mental).

In this connection particularly important coherence meditation ingredients seem to be “intentional consciousness, will, relaxation, focus, respiratory sinus arrhythmia, positive or renewing emotion, resilience, persistence, mantra, belief system, attitude, concentration, holistic orientation, meditation, contemplation, prayer, culture. Witnessing, biofeedback mechanism, action, good works” (68 Global Mental). Intentionality seems pre-eminent. For example, a related journal entry reads as follows:

“Intentionality or intentional act invoking directed consciousness. SHISO meditation with heart-breath devoted to each phase: spirit, heart, image (of heart), soul (as light streaming from or through point of nothingness at enter of heart just like global coherence app, then oneness of spirit enveloping all. Also used Lord’s Prayer, family prayer, and Christian Trinity mantra, the latter automatic, unconscious, with most heart-breath practice sessions” (80 Global Mental).

The following journal entries elaborate on meditation at physical, mental and spiritual levels respectively. In passing it should be noted that although distinctions such as one pointed versus objectless awareness may have academic, theoretical value in meditation research, practise may often involve a seamless slipping from one to the other, as reflected in the following examples.

“The Global Coherence app physically resembles a coloured, moving, *Trāṭaka* spiral (82 Global Physical), which evokes the image of a sacred planet in spirit motion towards an ultimate omega point where body, mind, spirit, personal, social and global coherence are all synchronized. (77 Global Mental). In addition to Hindu philosophical, meditation and yogic practices, which use *Trāṭaka* techniques [52] de Chardin’s Christian orientated theory of Centrology [44] could serve as an alternative theoretical foundation for further research into the Global Coherence app, whose colours may be perceived as moving inward to outward from Merton’s [53]) *le point vierge* or point of pure nothingness at heart centre, and outward to inward from spirit stars to heart (78 Global Mental)”.

“Practising radiating love meditation with personal mantra ‘Let us love the love that’s been loving us since the beginning now and forever:’ a word for each heart beat in concentration meditation. Knowledge that resting heart beat rhythm or pulse is 50 beats per minute approximately. This was an eyes closed session in keeping with phenomenology of using imaginative variation for meditation postures; e.g., eyes open or closed, sitting, lying, longer and shorter periods of time; as well as imaginative variation in meditation techniques, and comfort with Ultimate Reality, God, Brahman, *Shunyata*, Tao, Godhead, Allah, of most wisdom tradition practices, e.g., Hinduism, Taoism, Buddhism, Christianity, Islam. As Tao says, the name that can be named is not the eternal Name. From a human scientific point of view, further, greater praxis, integrity and sincerity will and should always yield greater or truer Truth” (83 Global Mental).

“Relaxing and releasing radiant love, whenever losing coherence and not sending love to environment and immediate others, as observed by less colour and movement of group coherence ring circling individual ring. This relaxing, releasing and radiating love is the essence of personal, social and global coherence practice. Also, yogic posture of *savasana*—lying flat on back, head propped up by pillow and another pillow on chest to prevent sensor touching skin and holding IPhone sideways enhances the relaxed concentrated effect. Whenever losing coherence, returning to the gesture of relax, release and radiate, especially by relaxing jaw, shoulder and chest muscles, so that body armour effect (clench into relax) does not prevent radiation of love through direct immediate contact with Presence, God, greater Being, Tao, whatever or however that Reality is known” (97 Global Spiritual).

This latter meditation reflection resonates with Bourgeault’s [54] Centering Prayer contemplation of a visionary, imaginal world, intelligible universe or energy field, common to all perennial wisdom consciousness, for which quantum physics is only a latter day purveyor of wisdom. Moreover, perennial wisdom traditions typically converge on the vital role of the universal, human heart as organ of spiritual, social and psychological perception whose holographic intelligence intuitively discerns perennial virtues and values to light the way. For example, Sufi, Helminski [55] could be describing HeartMath intuition and global coherence research [10] in the following quote:
“We have subtle subconscious faculties we are not using. In addition to the limited analytic intellect is a vast realm of mind that includes psychic and extrasensory abilities; intuition; wisdom; a sense of unity; aesthetic, qualitative, and creative capacities. Though these faculties are many, we give them a single name with some justification because they are operating best when they are in concert. They comprise a mind, moreover in spontaneous connection to the cosmic mind. This total mind we call “heart”.”

### 3.3. Integrated Evaluation 

A coherence adaptation of Wilber’s [40] integral approach to truth claims and all quadrants all levels (AQAL) model had direct relevance in enhancing reflexivity as concerns integrated findings and evaluation. However it should be noted that this model is also a simple heuristic, instructional device to structure discussion. By their very nature, models such as AQAL are artificial representations of reality and the quadrants are essentially artificial and overlapping. The interdependent origination of phenomena and inextricably interrelated nature of truth claims will become apparent in the ensuing discussion.

As observed in Table 5, when viewed from a coherence perspective, the model is perceived to concern interior and exterior perspectives of the individual and collective. From an interior individual perspective, heuristic phenomenological research concerns the quality of personal coherence, for example as reflected in attentiveness to meditation’s heartfelt sensations, perceptions, apprehensions, intuitions and insights and faithfulness in representing these as experiential data. From an exterior individual perspective, empirical measures provide objective correspondent truth claims, whether collectively significant or insignificant. Peer review and the test of time will determine the level of social agreement, for example among peer reviewers. Similarly relating to the degree of statistical significance, which overlaps individual and collective quadrants, exterior collective global and systemic dimensions determine coherent correspondent truth.

This following paragraph reflects interior, individual and subjective dimensions of this study. Phenomenological studies are implicitly intentional and interventional [30]. Certainly subjective intentional consciousness during meditation will have been an influential factor in significant Global Coherence app findings, and this is also the intended function of the app. Although this is the subject of another article, a particular example of creativity stimulated through heart awareness, sensation, perception, emotion, learning, cognition, memory and intentional consciousness deserves passing mention, not least for purposes of transparent bias suspension. “…The HeartMath system is the most complete psychology I know for the following reasons. It includes study of psyche, consciousness, behaviour, experience, physiology, society, spirituality, clinical, educational, environmental, and developmental psychology…” (47 Personal Mental).

From an interior collective and intersubjective perspective, indigenous Zulu tradition places special emphasis on communal ancestral interconnectedness with the Creator (*Unkulunkulu*), and the generations to come [56]. Moreover the Zululand area in general and Zulu culture is particular is globally famous for various reasons, not least because of its indigenous healing. One category of Zulu indigenous practitioner, the divine healer or *isangoma,* is particularly well known for using intuition (*umbilini*) and ancestral consciousness (*isangoma sabalozi*) [56]. Many other peoples across the planet have traditionally lived in a way that honours life as a deeply interconnected whole. Perennial philosophy and wisdom traditions typically recognise a nondualist oneness, layered in various levels of consciousness and evolving from matter through mind to spirit [1,40]. Wisdom traditions also recognise the reverse process that miraculous evolutionary happenings occur through some prior process of involution or emanation. This can be interpreted in terms of Centrology and increasing complexity/consciousness [44], which has considerable contemporary support [8,57].

Representing exterior, individual perspective, the empirical quantitative findings were particularly interesting for the following reasons. First was the unexpected Inner Balance findings of significant correlations between the participant’s coherence and achievement scores for three of the six magnetometers as well as total coherence. Second was the insignificance of the correlations with the South African Bonamanzi site, which is nearest geographical site to the author’s home and close to his heart in meaning. Third were the mirror reflection findings of significant correlations, albeit negative, between coherence and GCI06, as well as GCI06 and GCITOT, using the Global Coherence app. Reflexivity reveals that although case study disadvantages may include subjectivity and generalizability issues, these disadvantages may also be considered strengths, particularly in phenomenological case studies, qualitatively subjective by their very nature, that promote uniqueness, flexibility and in depth exploration, which, in this study, served to illuminate and stimulate personal, social and global exploration and health promotion [32,33,34]. Implications include research on integral and differential relationships and relative contributions of consciousness, intention and quantum physics.

Representing exterior, objective collective perspectives, collaborative, international, well controlled empirical research has indicated significant effects of divine healing intention on random event generators [58,59]. This finding resonates with HeartMath research on intuition and energetics, which has provided significant electrophysiological evidence of intuition as a holographic, system-wide, energetic process involving a non-local realm beyond the space-time world, which is mediated by the heart, before the brain [60]. As science is a human, cultural invention, like categories of time and space, it is not unexpected that the rigorous, empirical, evidence-based research pioneered by the HeartMath Global Coherence Initiative tends to support some enduring cultural beliefs and practices of indigenous peoples, promoted over millennia by intuitions, insights, meditations and life experiences of the sages, indigenous healers, doctors, diviners, prophets, priests and counsellors.

Integrated findings and evaluation perspectives from the present empirical and experiential study point towards support for HeartMath’s general postulate as to the existence of dynamic, information processing exchanges between all living systems and the earth’s energetic/magnetic field, allowing encoded information to be communicated subconsciously non-locally and globally [19,61]. By extension, meditation research aligned with research on the science of interconnectivity [62] pursued by the HeartMath Global Coherence Initiative has similar implications. For example, the present integral, empirical and phenomenological investigation includes and transcends what is called the imaginal world in Sufi traditions or Christ consciousness, or heaven on earth, in Christian traditions. Depth psychological traditions refer to veils and layers of consciousness, including the unconscious and superconscious, in [13,28,31]. Integral philosophy includes pre-rational, rational and post-rational, as well as arbitrary categories such as the natural, human and spiritual sciences [29]. In this context, the exponential growth of interdisciplinary fields such as bio-electromagnetic and subtle energy medicine [17] and bio-field sciences [63] predict various further distinctions under a broad umbrella paradigm such as energy science.

Concerning the technology used in this study, the Global Coherence app was experienced as valuable in its facility to focus meditation at differential levels of coherence, i.e., personal, social and global. It still need considerable technological development to approximate the variety of Inner Balance functions. In this context, the pattern of normal scientific and humanitarian progress should be vigorously pursued. This could include migration of all present Inner Balance and emWave software and related apps to the Global Coherence app, which version could be used for further rigorous scientific research. At the same time, philanthropic developers with the required financial sponsorship and other relevant resources could also pause to consider further teleology and purpose. Existing sensors are relatively expensive for users to monitor their actual personal, social and global contributions. Smart phones are now available to many people on the planet. Research could focus on equipping this freely downloadable app with relatively inexpensive monitoring devices that would enable people owning any smart phone to be able to contribute immediately and directly to regional and global coherence and health, as consumers may do for personal health through using any free health app. As a local example, this initiative could be succinctly expressed in isiZulu as *masihambisana*, let us all be coherent together. Focused global health instructional material as well as mobilization of health promoting resources could accompany this initiative. If this route is pursued, mass beneficial transformations in global heart intelligence, consciousness and related action may far exceed currently predicted trajectories. The potential and possibilities of using smart phones (accessible to most people) to improve the world seem endless.

## 4. Conclusions

Evaluation of the present study in terms of perceived individual and collective, interior and exterior dimensions, indicate that qualitative findings, especially those experiences derived from practice with the global coherence pp, were valuable in complementing quantitative findings in facilitating focus on global, as well as African, particularly South African, group and regional context. In this regard, in addition to global coherence, there seem compelling moral and ethical reasons for further development and promotion of coherence and health in Africa, as this particular continent has been relatively economically underdeveloped, except in the context of colonization, whose negative effects included looting of indigenous resources, especially natural resources such as gold and diamonds, and human resources as in the nefarious slave trade, which continues to this day. Considerable, moral, coherent, health promotion remains to be done in this context, as well as other contexts and regions of the global village. Finally, integrative findings may be interpreted as providing considerable theoretical and practical support for the HeartMath mission, vision and effectiveness of developing and promoting personal, social and global coherence.

## Figures and Tables

**Figure 1 ijerph-16-01245-f001:**
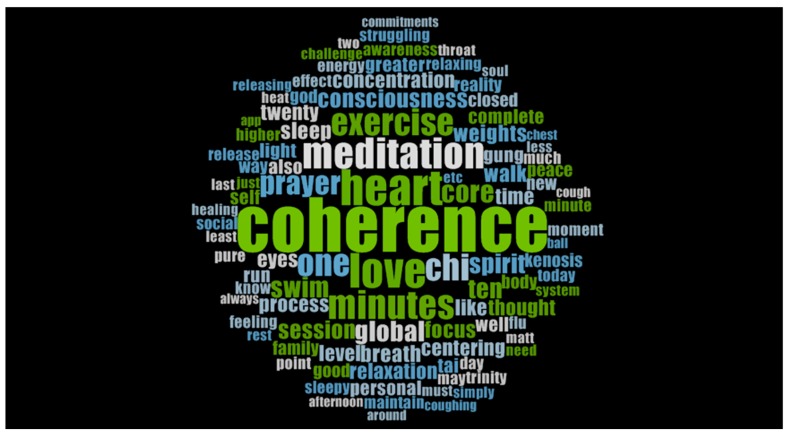
NVivo Word Cloud?

**Table 1 ijerph-16-01245-t001:** Descriptive Statistics for Inner Balance Coherence and Achievement scores.

Statistic	Coherence	Achievement
Mean	5.02	701.32
Number of sessions	153	153
Standard Deviation	0.94	477.12
Minimum	2.60	69.0
Maximum	7.40	1629.0
Range	4.80	1560
Sum	767.60	107,302

**Table 2 ijerph-16-01245-t002:** Inner Balance app and Global Coherence sites Spearmen correlation matrix.

	Cohere	Achieve	GCI01	GCI02	GCI03	GCI04	GCI05	GCI06	GCITOT
Cohere									
Achieve	−0.151								
GCI101	0.026	−0.143							
GCI102	−0.334 **	0.539 **	0.097						
GCI103	0.413 **	−0.655 **	0.131	−0.415 **					
GCI104	0.045	−0.117	0.361 **	0.174	0.211 **				
GCI105	−0.351 **	0.735 **	−0.046	0.632 **	−0.583 **	0.029			
GCI106	−0.020	−0.022	0.147	0.222 **	0.069	0.196 *	0.335 **		
GCITOT	−0.206 *	0.367 **	0.281 **	0.631 **	−0.220 **	0.325 **	0.733 **	0.777 **	

Cohere refers to average coherence, achieve to average achievement score. GCI001 California, USA; GCI002, Hofuf, Saudi Arabia; GCI003 Lithuania; GCI004 Alberta, Canada; GCI005 Northland, New Zealand; GCI006 Hluhluwe, South Africa. GCITOT: total global coherence.

**Table 3 ijerph-16-01245-t003:** Global Coherence App and Global Coherence sites Spearmen correlation matrix.

Site	Cohere	GCI01	GCI02	GCI03	GCI04	GCI05	GCI06	GCITOT
GCI101	0.107							
GCI102	−0.249	0.193						
GCI103	0.011	0.401 **	0.538 **					
GCI104	0.143	0.419 **	−0.023	0.379 **				
GCI105	−0.082	−0.334 **	0.278	0.423 **	−0.014			
GCI106	−0.305 *	0.012	0.304 *	0.322 **	−0.090	−0.061		
GCITOT	0.055	0.156	0.561 **	0.642 **	0.103	0.211	0.882 **	

Cohere refers to average coherence, achieve to average achievement score. GCI001 California, USA; GCI002, Hofuf, Saudi Arabia; GCI003 Lithuania; GCI004 Alberta, Canada; GCI005 Northland, New Zealand; GCI006 Hluhluwe, South Africa. GCITOT: total global coherence.

**Table 4 ijerph-16-01245-t004:** Integral consciousness model of physical, mental and spiritual dimensions of personal, social and global coherence.

Personal	Social	Global
Physical 5, 7, 8, 10, 12, 14, 19, 21, 22, 33, 39, 45, 46, 58, 59, 60, 62, 65, 67. (19 entries)	Physical 17, 24, 26 (3 entries)	Physical 43, 53, 55, 82. (4 entries)
Mental 1, 13, 30, 31, 32, 35, 36, 41, 47, 48, 49, 57, 61, 63, 64, 73, 75, 84, 87, 94. (20 entries)	Mental 3, 4, 11, 15, 27, 28, 29, 34, 50, 52, 69. (11 entries)	Mental 23, 25, 68, 76, 77, 78, 80, 81, 83, 85, 86, 92, 96, 100. (14 entries)
Spiritual 16, 37, 38, 56, 71, 74, 79, 88, 89, 95. (11 entries)	Spiritual 18, 54, 72, 98, 99. (5 entries)	Spiritual 2, 6, 9, 20, 40, 42, 44, 51, 66, 70, 90, 91, 97. (13 entries)

**Table 5 ijerph-16-01245-t005:** Integral Coherence Model of Truth Claims.

	Interior	Exterior
Individual	Personal coherence, Authenticity, faithfulness Subjective truthfulness	Individual coherence objective truth towards significant or insignificant correlations.
Collective	Social coherence Agreement amongst experts Peer review	Global coherence Alignment with data and findings from other systems and studies.

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
