# Peer review of "Empirical and Heuristic Phenomenological Case Study of the HeartMath Global Coherence Initiative"

_ijerph, 2019, doi:10.3390/ijerph16071245_

Round 1
Reviewer 1 Report
The study is current and of immense interest in present day happenings around the world. The introduction, as well as methodology is well set out and clearly delineated. The turbos is an expert in this field and it is evident in the flow of the arguments raised. It would be great to see a study using a larger sample.
References to McCraty et al (26) - should it not be written out in full first time or does it refer to either (line 79 - 86) another reference?
Author Response
Thanks for referencing comments. In response to "References to McCraty et al (26) - should it not be written out in full first time or does it refer to either (line 79 - 86) another reference?"
According to APA Publication Manual Six Edition, if a work has six (6) or more authors, cite only the last name of the first author followed by et al. each time you refer to this work. Therefore paragraph lines 79-86 has been modified as follows:
In this context, health is understood in terms of diverse arrays of dynamic energy including physiological systems, behavioural patterns and environmental resonant frequencies [19]. Various environmental health studies provide examples. McCraty, et al. [23] found significant HRV correlations with solar and geomagnetic activity as well as geomagnetic field-line and Schumann resonances over a longitudinal 31-day period in a group of participants in separate locations. Timofejeva, et al. [24] found synchronization between slow HRV wave rhythms and changes in local magnetic field data, with the degree of synchronization being associated with the quality of interpersonal relationships. Extending earlier studies, Alabdulgader, et al. [25] as well as McCraty et al., [26] again indicated that energetic environmental phenomena impact psychophysical processes in people in different ways depending on their sensitivity, health status and capacity for self-regulation.
Reviewer 2 Report
The author is a semi-retired 69 year old with much thoughtful multicultural experience. As Alex Haley once wrote: “the death of an elder is like the burning of a library”. Since I am a decade older with parallel multicultural academic and research experience I will attempt the temerity to approach this comprehensive work from a critical perspective. Both our libraries remain open. On the positive side, this is a fine example of a well referenced and explained yet alternative theoretical viewpoint with the case example of the author’s single outcomes as illustration more than conclusive research. The author acknowledges this and recommends further more controlled research with larger numbers of participants. The wide range of important philosophical and psychological traditions within an important primary (and rare) South African cultural perspective makes this an important international contribution to the journal.
Some other comments or suggestions though might include: - The sentences are often very long, comprising four or more lines. One rule of thumb I like to suggest is that if you read a sentence out loud and run out of breath before completing it, it needs shortening.
–The categorization of qualitative data categories may be given more credibility if another experienced analysis is done, especially by a non-participant non-author. Such validation, given equal or close categorical conclusions may reduce some of the limitations. -In the abstract and again elsewhere in the text, the null hypothesis “is indicated”. In fact a null hypothesis can never be proved or indicated. Research can only summon enough evidence to reject it, or without that sufficient evidence in this one study, fail to reject it (yet). -In the body of the text, references are indicated by {number} in medical research tradition but not by APA or general publication practice as (author, year). This journal’s editorial practice may be flexible in this regard? -The discussion is particularly valuable in another very thorough piece illustrating some study-wide well written material integrating controversial ancient perspectives with modern applications. The refreshing suggestions for modern smart phones and their apps allowing for further contemporary research and usage is such an example.. Overall well worth publishing this challenging piece.
Author Response
Response to the Second Review - Empirical and Heuristic Phenomenological Case Study of the HeartMath Global Coherence Initiative
The meaningful, valuable review by the second reviewer is deeply appreciated
In response to the comment on long sentences, the following line changes have been made to the revised document:
164-165. The author’s involvement in this study may be understood in terms of three historical streams of evidence. The first.
206-207. and report readiness after 64 recorded sessions in 38 days coherence practice with the app. At this stage the author’s individual coherence points recorded were 59,221; the group coherence points
355-359. It needs passing mention that the philosophical and poetic quotes reflect both the author’s inclinations as well as their propensity to describe, amplify and elaborate experiential meaning. This is required such qualitative research criteria as truthfulness, credibility, faithfulness, integrity, accuracy, and dependability, in experiential and scientific fields with no final answers.
360-363. Holistic healing consciousness. For want of a better one, the tautological phrase “holistic healing” is used to convey the ongoing experience of the whole, or of healing as whole making event, occurring moment to moment. This is also an evolutionary process, connecting continually towards greater wholeness as distinct from simultaneous processes of entropy and dissolution.
544-549. For example, the present integral, empirical and phenomenological investigation includes and transcends what is called the imaginal world in Sufi traditions or Christ consciousness, or heaven on earth, in Christian traditions. Depth psychological traditions refer to veils and layers of consciousness, including the unconscious and superconscious. [13, 28, 31]. Integral philosophy includes pre-rational, rational and post-rational, as well as arbitrary categories such as the natural, human and spiritual sciences [29].
In response to the comment “The categorization of qualitative data categories may be given more credibility if another experienced analysis is done, especially by a non-participant non-author. Such validation, given equal or close categorical conclusions may reduce some of the limitations.
In fact the first course sieve NVivo did take place in the company of a non-participant, non-author, who was also using NVivo technology with his research. In the second level or phase involving allocation of 100 responses to one of 9 dimensions in an integral theory driven table, ninety nine percent agreement was reached with a second collaborative researcher.
In response to the comment -In the abstract and again elsewhere in the text, the null hypothesis “is indicated”. In fact a null hypothesis can never be proved or indicated.
This comment is very well taken and one the author of the present submission has often made in reviews. In this instance there may have been a misunderstanding with regard to the use of the word “indicated” in the abstract. Consequently the sentence in question (line 11-12) has been changed to read as follows: “As this was a single case study, with a limited amount of data, the null hypothesis was invoked. The word invoked is also used in line 93.
In response to the comment: “In the body of the text, references are indicated by {number} in medical research tradition but not by APA or general publication practice as (author, year). This journal’s editorial practice may be flexible in this regard? “
The original APA style in which the article was submitted has already been modified as required by the Assistant Editor IJERPH